# A Comprehensive Study of Artificial Intelligence Applications for Soil Temperature Prediction in Ordinary Climate Conditions and Extremely Hot Events

**Hanifeh Imanian** [1,*] **, Juan Hiedra Cobo** [2] **, Pierre Payeur** [3] **, Hamidreza Shirkhani** [2]
**and Abdolmajid Mohammadian** [1]

1   School of Electrical Engineering and Computer Science, University of Ottawa, Ottawa, ON K1N 6N5, Canada;
    majid.mohammadian@uottawa.ca
2   National Research Council Canada, Ottawa, ON K1A 0R6, Canada; juan.hiedracobo@nrc-cnrc.gc.ca (J.H.C.);
    hamidreza.shirkhani@nrc-cnrc.gc.ca (H.S.)
3   Computer Science Engineering Department, University of Ottawa, Ottawa, ON K1N 6N5, Canada;
    ppayeur@uottawa.ca
*   Correspondence: himania3@uottawa.ca

**Abstract:** Soil temperature is a fundamental parameter in water resources and irrigation engineering. A cost-effective model that can accurately forecast soil temperature is urgently needed. Recently, many studies have applied artificial intelligence (AI) at both surface and underground levels for soil temperature predictions. In the present study, attempts are made to deliver a comprehensive and detailed assessment of the performance of a wide range of AI approaches in soil temperature prediction. In this regard, thirteen approaches, from classic regressions to well-established methods of random forest and gradient boosting to more advanced AI techniques, such as multi-layer perceptron and deep learning, are taken into account. Meanwhile, great varieties of land and atmospheric variables are applied as model inputs. A sensitivity analysis was conducted on input climate variables to determine the importance of each variable in predicting soil temperature. This examination reduced the number of input variables from 8 to 7, which decreased the simulation load. Additionally, this showed that air temperature and solar radiation play the most important roles in soil temperature prediction, while precipitation can be neglected in forecast AI models. The comparison of soil temperature predicted by different AI models showed that deep learning demonstrated the best performance with R-squared of 0.980 and NRMSE of 2.237%, followed by multi-layer perceptron with R-squared of 0.980 and NRMSE of 2.266%. In addition, the performance of developed AI models was evaluated in extremely hot events since heat warnings are essential to protect lives and properties. The assessment showed that deep learning and multi-layer perceptron methods still have the best prediction. However, their R-squared decreased to 0.862 and 0.859, and NRMSE increased to 6.519% and 6.601%, respectively.

**Keywords:** artificial intelligence; climate prediction; deep learning; extreme heat events; multi-layer perceptron; neural network; regression; soil temperature



## 1. Introduction

Soil temperature is a pivotal parameter in geo-environmental and geotechnical engineering. Soil temperature prediction is significant for atmospheric models, numerical hydrological and land-surface hydrological processes, as well as land–atmosphere interactions [1,2]. In addition, in some other fields, such as water resources and hydrologic engineering, soil temperature is an important factor [3]. Soil temperature is a catalyst for many biological processes. It influences the soil moisture content, aeration and availability of plant nutrients, which are necessary for plant growth. It is essential to measure or estimate this parameter with a reasonable precision. Therefore, an accurate and cost-effective model that can accurately predict soil temperature is urgently needed [4–6].

There are two common ways of obtaining soil temperature: direct measurement and indirect prediction using numerical models [7]. Since soil temperature is a stochastic parameter and similar to other climatic parameters, researchers use the following approaches to calculate it: statistical models and machine learning methods [8].

Statistical models use historical time series to estimate soil temperature predictions. The commonly used method for time series forecasting is stochastic modeling, such as the auto-regressive moving average (ARMA) and auto-regressive integrated moving average (ARIMA) [9]. These statistical methods assume that changes in the statistical properties of soil temperature data series in the future would be similar to those in the past. This means that large amounts of data are required for long-term predictions. Bonakdari et al. (2019) and Zeynoddin et al. (2020) proposed a linear stochastic method to model daily soil temperature with sufficient knowledge of the time series structure [6,9].

Recently, the use of artificial intelligence (AI)-based techniques for predicting real-world problems has rapidly increased. Many studies applied AI models at both surface and under-ground levels for soil temperature predictions. George (2001) made use of a multi-layer neural network for a weekly mean soil temperature prediction during 1 year [10]. Monthly soil temperature was modeled using a 3-layer artificial neural network (ANN) constructed by Bilgili (2010) [11]. He used meteorological variables of atmospheric temperature, atmospheric pressure, relative humidity, wind speed, rainfall, global solar radiation and sunshine duration to make predictions at five depths below ground level and compared them with linear and nonlinear regression results. Ozturk et al. (2011) developed feed-forward artificial neural network models to estimate monthly mean soil temperature at five depths from 5 to 100 cm under the ground using meteorological data such as solar radiation, monthly sunshine duration and monthly mean air temperature [12]. Zare Abyaneh et al. (2016) used ANNs and a co-active neuro-fuzzy inference system for the estimation of daily soil temperatures at six depths from 5 to 100 cm underground using only mean air temperature data from a 14-year period as input data [13]. An adaptive neuro-fuzzy inference system (ANFIS), multiple linear regression (MLR) and ANN models were developed by Citakoglu (2017) to predict soil temperature data in monthly units at five depths from 5 to 100 cm below the soil surface using monthly air temperatures and monthly precipitation for at least 20 years [14]. Himika et al. (2018) made use of various existing regression and machine learning models to propose an ensemble approach to predict land temperature [15]. The chosen models were decision tree, variable ridge regression and conditional inference tree. Delbari et al. (2019) evaluated the performance of a support vector regression (SVR)-based model in estimating daily soil temperature at 10, 30 and 100 cm depth at different climate conditions [16]. Climatic data used as inputs for the models were air temperature, solar radiation, relative humidity, dew point, and the atmospheric pressure. They compared the obtained results with classical MLR and found that SVR performed better in estimating soil temperature at deeper layers. A study by Alizamir et al. (2020) compared four machine learning techniques, extreme learning machine (ELM), artificial neural networks (ANN), classification and regression trees and a group method of data handling in estimating monthly soil temperatures [3]. They used monthly climatic data of air temperature, relative humidity, solar radiation, and windspeed at four different depths of 5 to 100 cm as model inputs. ELM was found to generally perform better than the others in estimating monthly soil temperatures. Li et al. (2020) presented a novel scheme for forecasting the hourly soil temperature at five different soil depths [17]. They developed an integrated deep bidirectional long short-term memory network (BiLSTM) and fed their model with air temperature, wind speed, solar radiation, relative humidity, vapor pressure and dew point. Six benchmark algorithms were chosen to prove the relative advantages of the proposed method, namely, three deep learning methods: LSTM, BiLSTM and deep neural network (DNN), and three traditional machine learning methods: random forest (RF), SVR, and linear regression. The proposed model of Penghui et al. (2020) is a hybridization of an adaptive neuro-fuzzy inference system with optimization methods using a mutation salp swarm algorithm and grasshopper optimization algorithm (ANFIS-mSG) [18]. The pre-

diction of daily soil temperatures was conducted based on maximum, mean and minimum air temperature. The results are compared with seven models, including classical ANFIS, a hybridized ANFIS model with grasshopper optimization algorithm (GOA), salp swarm algorithm (SSA), grey wolf optimizer (GWO), particle swarm optimization (PSO), genetic algorithm (GA), and dragonfly algorithm (DA). Shamshirband et al. (2020) modeled air temperature, relative humidity, sunshine hours and wind speed using a multilayer perceptron (MLP) algorithm and SVM in hybrid form with the firefly optimization algorithm (FFA) to estimate soil temperature at 5, 10 and 20 cm depths [19]. In a study by Seifi et al. (2021), hourly soil temperatures at 5, 10, and 30 cm depths were predicted by applying ANFIS, SVM, MLP and a radial basis function neural network with optimization algorithms of SSA, PSO, FFA and sunflower optimization (SFO) [4]. They used air temperature, relative humidity, wind speed and solar radiation as input information and found that wind speed did not have a high coherence with soil temperature. A generalized likelihood uncertainty estimation approach was implemented to quantify model uncertainty and concluded that ANFIS-SFO produced the most accurate performance. Hao et al. (2021) proposed a model called a convolutional neural network based on ensemble empirical mode decomposition (EEMD-CNN) to predict soil temperatures at three depths between 5 and 30 cm [1]. They used the statistical properties of the maximum, mean, minimum and variance air temperatures as the meteorological input information. The results were compared using four models: persistence forecast (PF), backpropagation neural network, LSTM and EEMD-LSTM. In a similar study, a convolutional 3D deep learning model with ensemble empirical mode decomposition (EEMD) was proposed by Yu et al. (2021) to predict soil temperatures over 1, 3 and 5 days at a depth of 7 cm underground [2].

　　　Extreme events and their related irregular data are observed in many climate time-series. So, the development of efficient methods to understand and accurately predict such representative features remains a big challenge. O'Gorman and Dwyer (2018) applied the RF method to simulate extreme precipitation events [20]. Hu and Ayyub (2019) used machine learning to propose an alternative method for extreme precipitation modeling [21]. A deep learning strategy is proposed by Qi and Majda (2020) to predict the extreme events that appear in turbulent dynamical systems [22]. Huang et al. (2021) employed a stacking model and the XGBoost model to study the trend analysis between extreme land surface temperatures and the amount of solar radiation [23]. Araújo et al. (2022) propose an approach based on networks to forecast extreme rainfall [24]. In addition, Bochenek and Ustrnul (2022) reviewed 500 publications from 2018 to 2021 concerning machine learning methods, finding that the phrase "extreme events" was recorded a few times in the field of climate and numerical weather predictions, mostly used in terms of extreme precipitation and streamflow [25]. So, the ability of different AI strategies in the prediction of extreme events should be investigated.

　　　The above literature review shows that there are some gaps in the knowledge of AI applications in the prediction of soil temperature. First, there is an absence of a comprehensive and detailed assessment of the performance of different artificial intelligence approaches, from linear regression to complicated advanced techniques in soil temperature estimation. Second, the most important atmospheric variables to be used as input data for AI models must be determined. Previous studies usually used limited atmospheric variables, while in the current investigation, a wide range of variables were employed. Although several researchers developed codes equipped with some AI models, they focused on limited meteorological parameters, mostly air temperature. There are many other climate data that affect soil temperature, directly or indirectly. Therefore, the impact of other land and atmospheric variables needs to be further studied. Furthermore, research should evaluate and demonstrate the AI models' skill in successfully tackling extreme events.

　　　The main purpose of this study is to evaluate the performance of a wide range of AI approaches to soil temperature prediction using various land and atmospheric variables. In this article, 13 methods, from classic regressions, well-established methods of random forest and gradient boosting to advanced AI techniques, such as ANFIS, ANN and deep learning,

are taken into account. Meanwhile, a broad selection of variables from a comprehensive reanalysis of ERA5 datasets were chosen as input parameters for the developed prediction model to consider different aspects of the problem.

The rest of the paper is organized as follows: Section 2 describes the study area and involved parameters, introduces the applied AI approaches and reviews the methodology of study. The evaluation metrics are also presented in this section. The subsequent section defines datasets and input information, presents the results and compares them with the actual data. A discussion on the performance of the different AI methods using error metrics, confidence bands and a sensitivity analysis of the outcomes of input parameters is given in Section 4. Additionally, the behavior of developed AI models in extreme events is investigated in the same section. The last section presents the concluding remarks and future study suggestions.

## 2. Materials and Methods

### 2.1. Study Area and Dataset

The climate data used in the present study were obtained from ERA5. They were downloaded from the freely accessible website of Climate Data (https://cds.climate.copernicus.eu/, accessed on 1 July 2021). ERA5 is the fifth-generation atmospheric reanalysis of the global climate covering the period from 1950 to present. ERA5 is produced by the Copernicus Climate Change Service (C3S) at ECMWF. It provides hourly estimates of a large number of atmospheric, land and oceanic climate variables in a gridded-base format with a regular latitude–longitude grid. The data coverage is global with a horizontal resolution of $0.25° \times 0.25°$ and resolves the atmosphere using 137 levels from the surface up to a height of 80 km. ERA5 includes information about uncertainties for all variables at reduced spatial and temporal resolutions. ERA5 combines vast amounts of historical observations into global estimates using advanced modeling and data assimilation systems.

The study area is Ottawa, the capital city of Canada (45.4° N, 75.7° W), located in the southeast of the country, in the province of Ontario. Figure 1a shows the geographical location of the considered site used in this study [26]. Ottawa has a semi-continental climate with four distinct seasons. It has a warm, humid summer and a very cold and harsh winter.

Six stations represent the city of Ottawa, which are specified with red circles in Figure 1b. The stations cover an area of approximately $30 \times 40$ km. The coordinates of the stations are shown in Table 1.

**Table 1.** Coordinates of considered stations in city of Ottawa.

| Station No. | Latitude | Longitude | Station No. | Latitude | Longitude |
|:---:|:---:|:---:|:---:|:---:|:---:|
| #1 | 45.25° N | 75.50° W | #4 | 45.50° N | 75.50° W |
| #2 | 45.25° N | 75.75° W | #5 | 45.50° N | 75.75° W |
| #3 | 45.25° N | 76.00° W | #6 | 45.50° N | 76.00° W |

The used variables were the hourly weather conditions, including air temperature at 2 m above the surface (Kelvin), total precipitation (m), surface pressure (Pa), evaporation (m of water), instantaneous wind gusts at 10 m above the surface (m/s), dewpoint temperature 2 m above the surface (Kelvin), surface net solar radiation ($J/m^2$) and surface net thermal radiation ($J/m^2$). The valid data were collected from 1 June to 31 August 2020, a total of 92 days [27]. The data from these variables in the six mentioned stations are considered as model inputs. Then, approximately 106,000 pieces of climatic information were gathered as the AI model's input. The output of each AI model predicted hourly soil temperature in Kelvin at the layer between 0 to 7 cm underground.

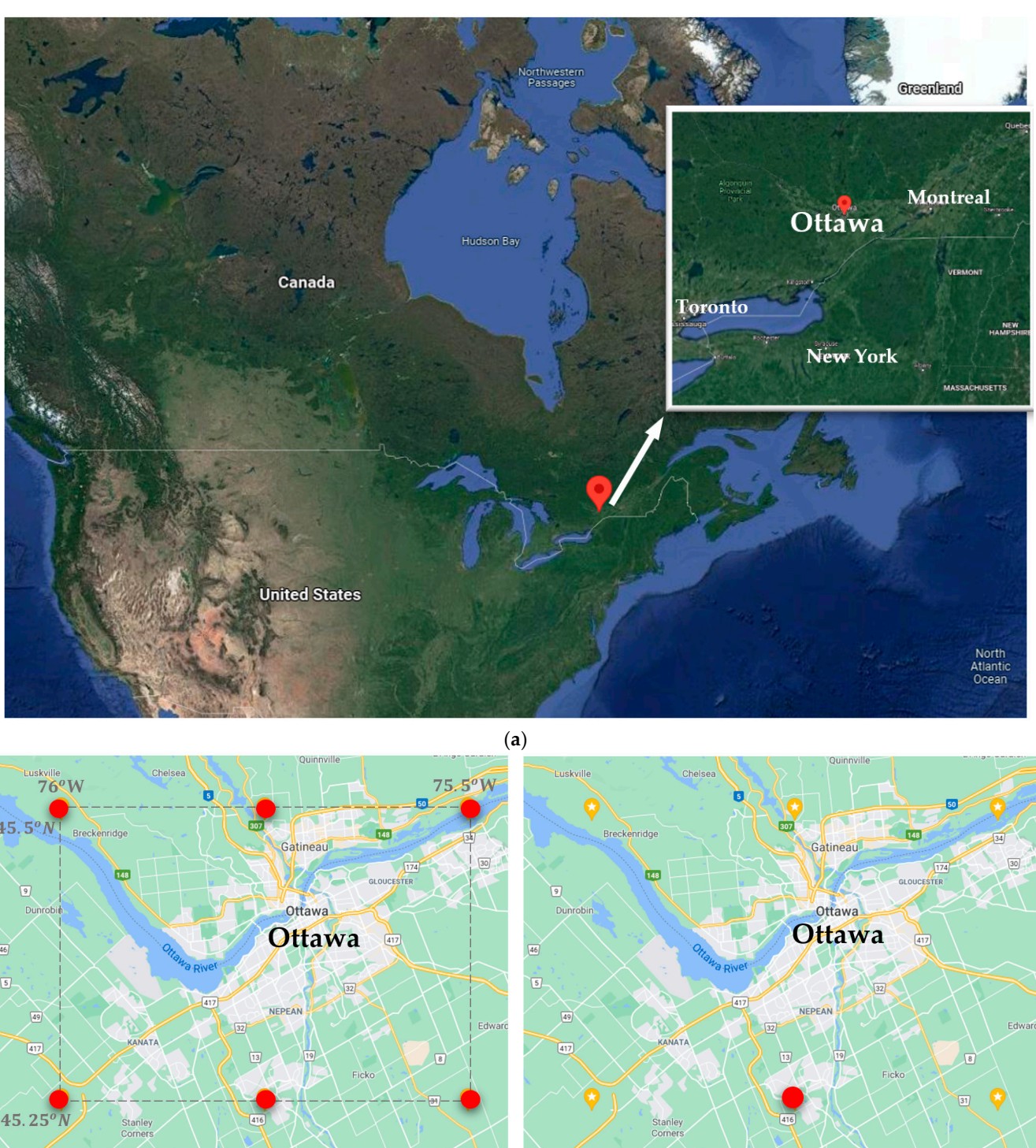

**Figure 1.** Location of (**a**) the site (Ottawa) considered in the present study, (**b**) six stations representing Ottawa, (**c**) one selected station.

## 2.2. Descriptions of Artificial Intelligence Algorithms

A wide range of AI approaches are applied in the developed numerical model, as described below.

There are two ways to assign hyperparameters in the models: Some of the hyperparameters are obtained by sensitivity analysis. In this technique, different values for hyperparameters are assumed to be in the acceptable range. Then, the model is executed, and its behavior is monitored. Finally, the value with the best outcome is chosen for that

hyperparameter. In the other method, hyperparameters are set to a typical value according to the previously studied cases or other considerations.

Most of the hyperparameters in the current study were assigned after a sensitivity analysis, while some activation functions or kernel functions were set to recommend typical functions according to the present studied case, which is regression.

### 2.2.1. Linear Regression, Ridge, Lasso and Elastic Net

Four different linear models were applied in the developed code, including: Linear regression, ridge, lasso and elastic net. Linear regression is the most basic form of a linear model for minimizing the residual sum of squares. So, the objective function is as follows [11,14,17,28]:

$$\sum(y - Xw)^2 \tag{1}$$

where $y$ is the actual value, $X$ is the input value and $w$ is weight.

Ridge is a linear model that imposes a penalty on the sum of squared value of the weights, resulting in a group of weights that are more evenly distributed. The objective function called L2 regularization:

$$\sum(y - Xw)^2 + \alpha \sum w^2 \tag{2}$$

where $\alpha$ is a non-negative hyperparameter that controls the magnitude of the penalty. In the present study, $\alpha$ was set to 0.1 by a sensitivity analysis of the ridge model.

For lasso, a modification of linear regression is applied, in which the model is penalized for the sum of absolute values of the weights, which is known as L1 regularization:

$$\frac{1}{2m}\sum(y - Xw)^2 + \alpha \sum |w| \tag{3}$$

In the present study, $\alpha$ was set to 0.001 by sensitivity analysis for lasso model.

Elastic net (Enet) is a combination of the two last models such that both regularizations related to ridge and lasso models are exerted on the linear regression. Parameter $\rho$ defines the ratio of penalties. If $\rho$ is zero, the penalty would be L2 regularization; if $\rho$ is one, the penalty would be L1 regularization:

$$\frac{1}{2m}\sum(y - Xw)^2 + \alpha * \rho \sum |w| + \frac{\alpha}{2}(1 - \rho) * \sum w^2 \tag{4}$$

In the present study, $\rho$ was set to 0.5, so it was a combination of ridge and lasso models. Additionally, $\alpha$ was set to 0.0001, obtained by a sensitivity analysis for Enet model.

To reach the desired hyperparameters, different values in the acceptable range are usually tried, and the one that leads to the best result with minimum error is chosen. However, the intermediate process is not explained in the manuscript to avoid prolonging the text, and only the selected value for hyperparameter was provided. For example, in the Enet method, different values of $\alpha$ and related error indicators are presented in Table 2. At the beginning, it can be seen that the error metrics improved, but with the decreasing $\alpha$ from 0.0001, the error indicators did not dramatically change. So, this value was selected for hyperparameter $\alpha$ in this method.

**Table 2.** Results of sensitivity analysis of hyperparameter $\alpha$ in Enet model.

| Hyperparameter $\alpha$ | 1E-2 | 5E-3 | 1E-3 | 5E-4 | 2E-4 | 1E-4 | 1E-5 | 1E-10 |
|---|---|---|---|---|---|---|---|---|
| NRMSE | 0.04807 | 0.04693 | 0.04628 | 0.04623 | 0.04620 | 0.04619 | 0.04618 | 0.04618 |
| $R^2$ | 93.95% | 94.23% | 94.39% | 94.40% | 94.41% | 94.41% | 94.41% | 94.41% |
| NRMSE difference | - | −2.43% | −1.40% | −0.11% | −0.06% | −0.02% | −0.02% | 0.00% |
| $R^2$ difference | - | 0.30% | 0.17% | 0.01% | 0.01% | 0.00% | 0.00% | 0.00% |

The change rate is calculated in the last two rows of the table. It shows NRMSE decreases fast with greater $\alpha$ values, but it becomes almost constant when $\alpha$ is selected as 0.0001 and less. The same trend can be seen for R-squared. While $\alpha$ values are high, R-squared increases and reaches a point that remains unchanged for $\alpha$ equals 0.0001. This analysis confirms that 0.0001 is a good choice for the $\alpha$ hyperparameter in the Enet model since larger values lead to high error results and do not show an acceptable accuracy. Additionally, lower values do not significantly improve model results.

### 2.2.2. Nearest Neighbors Method

In the nearest neighbors method, learning is based on the fixed number of nearest neighbors for each query point or on the neighbors within a fixed radius of the query point. It can be uniformly weighted or made proportional to distance.

The present study implemented learning for k-nearest neighbors' points (KNN). Parameter k was set to 3 after calibration. Additionally, a uniform weight function was used in the prediction, which means each point in the local neighborhood uniformly contributed to the classification of a query point. The distance function used for the tree was Euclidean distance, which is the ordinary distance between two points on a plane.

### 2.2.3. Decision Tree, Random Forest, Gradient Boosting and Extreme Gradient Boosting

There is another learning method called decision trees. This method's goal is to create a model that predicts the value of a target variable by learning simple decision rules inferred from data features. In other words, a tree can be seen as a piecewise constant approximation.

Ensemble learning method is a technique that combines predictions from multiple machine learning algorithms in order to make a more accurate prediction than a single model. Some ensemble models are developed based on decision trees such as random forest and gradient boosting.

Random forest is a meta estimator that fits a number of decision trees on various subsets of the dataset. A RF operates by constructing several decision trees during training time and outputting the mean of the classes as the prediction for all the trees. Several trees run in parallel with no interactions amongst them in this method [7]. In the present study, the number of decision trees in the random forest model was set to 1000 after sensitivity analysis. The criterion function to measure the quality of a split in RF was squared error since other functions such as absolute error are significantly time-consuming. No maximum depth was assigned for the tree, so nodes could expand until all leaves became pure.

Gradient boosting repeatedly fits a decision tree on the differentiable loss functions. This method builds one decision tree at a time, where each new tree helps to correct errors made by the previously trained tree. Gradient boosting is fairly robust against over-fitting, so a large number of boosting stages usually results in a better performance. In the present study, the number of boosting stages to perform was set to 1000 after sensitivity analysis. More repetition did not significantly improve the results.

Extreme gradient boosting (XG boost) builds a model by a set of trees, reduces the errors, and builds a new model in subsequent iterations. Unlike the gradient boosting method, XG boost implements some regularization; therefore, it helps to reduce overfitting. Additionally, it is much faster compared to gradient boosting [29,30].

There are some hyperparameters that should be tuned for the XG boost model: (i) number of used decision trees, which would often be better if have more, (ii) tree depth, which controls how specialized each tree is to the training dataset, (iii) learning rate, which controls the amount of contribution of each decision tree model on the ensemble prediction. In the present study, the number of decision trees, tree depth and learning rate were set to 100, 6 and 0.3, respectively.

### 2.2.4. Support Vector Machine

The basic idea of support vector machines is to map the original data into a feature space with a high dimensionality through a non-linear mapping function and construct an optimal hyperplane in the new space. In case of regression, a hyperplane is constructed that lies close to as many points as possible. It means the optimal hyperplane it seeks is not to maximize the separation distance between two or more kinds of sample points such as SVM, but to minimize the total deviation between sample points and the hyperplane [31–34].

When solving nonlinear problems, SVR applies the kernel function to map the nonlinear regression problem to the space of higher latitude, so that an optimal hyperparameter to be constructed that lies close to as many points as possible.

The radial basis function kernel is widely used for SVM models and recommended in regression problems [32] and was applied in this study:

$$K(X_i, X_j) = exp\left(-\gamma \sum (X_i - X_j)^2\right) \tag{5}$$

where $X_i$, $X_j$ are two points, and the hyperparameter $\gamma$ passes the reciprocal of the number of features, which was set to 8 in the present study.

### 2.2.5. Stacking Method

Another group of methods is called stacking or stacked generalization. Stacking is an ensemble machine learning algorithm that learns how to best combine the predictions from multiple high-performance machine learning models. In this paradigm, the outputs of some aforementioned individual estimators are gathered and an additional regressor is used to compute the final prediction. Stacking often ends up with a model that is better than any individual intermediate model [35].

In the present study, different combinations of estimators are tried, and eventually, three methods—random forest, SVM and ridge—computed the best outcome.

In the first step, the stacking method uses some predictors, which consist of a list of machine learning methods stacked together in parallel on the input data. Herein, the random forest and SVM methods were applied as predictors. At the second step of stacking method, a final predictor is employed that uses the predictions of the first estimators as inputs. The final predictor is a machine learning regressor, which was chosen to be the ridge method in the current study.

### 2.2.6. Multi-Layer Perceptron

Multi-layer perceptron (MLP), a class of feedforward ANN, is a non-linear function approximator in layers using back propagation with no activation function in the output layer. It used the rectified linear unit (Relu) function as the activation function in the hidden layers [7,8,36–39]:

$$g(z) = max(0, z) \tag{6}$$

Although there are some different types of activation functions used in neural networks, Relu is the most common function for hidden layers because it is both simple to implement and effective at overcoming the limitations of other previously popular activation functions.

MLP uses different loss functions depending on the problem type. For the case of prediction, MLP uses the square error loss function; written as:

$$\frac{1}{2} \sum (y - Xw)^2 + \frac{\alpha}{2} \sum w^2 \tag{7}$$

Starting from initial random weights, MLP minimizes the loss function by repeatedly updating these weights. After computing the loss, a backward pass propagates it from the output layer to the previous layers, providing each weight parameter with an update

value meant to decrease the loss. The algorithm stops when it reaches a pre-set maximum number of iterations, or when the improvement in loss is below a certain, small number.

In the current study, 700 hidden layers were used in the MLP model after sensitivity analysis. The sensitivity analysis showed that using 500 and 1000 hidden layers led to NRMSE of 0.04050 and 0.03655, respectively. Whereas, considering 700 hidden layers, NRMSE was 0.03635, which showed an improvement in error metrics.

The activation function for the hidden layers was Relu, which passes the maximum of the variable and zero. The solver for weight optimization was Adam, which is a stochastic gradient-based optimizer. Maximum number of iterations was 5000, which determines the number of epochs, meaning how many times each data point is used.

### 2.2.7. Deep Learning

Briefly, deep learning is a machine learning technique that employs a deep neural network. The word "deep" refers to the depth of layers in a neural network. A deep neural network is a multi-layer neural network that contains two or more hidden layers. However, it is not just the addition of hidden layers or the addition of nodes in the hidden layer. There is no point in adding hidden layers if they cannot be trained, and the neural network with deeper layers may not be appropriately trained [17,37]. Although deep learning shows outstanding achievements, it does not actually have any critical technologies to present. The innovation of deep learning is a result of many minor technical improvements. Technically, the backpropagation algorithm experiences three primary difficulties in the training process of the deep neural network, and deep learning could overcome those problems [40]:

-   Vanishing gradient: This is a problem when the hidden layers are not adequately trained. Deep learning is assisted by some numerical method that better achieves the optimum value and is beneficial for the training of the deep neural network.
-   Overfitting: Deep neural networks are vulnerable to overfitting because the model becomes more complicated as it includes more hidden layers, and hence more weight. Deep learning solves this problem by training only some of the randomly selected nodes rather than the entire network. It also uses regularization.
-   Computational load: Deep learning relieved significant training time due to heavy calculations by employing GPU and some other algorithms.

In the deep learning method, the number of epochs is the number of times that the entire training dataset is shown to the network during training. In the current study, this hyperparameter was set to 100. The optimization algorithm used to train the network was Adam, which is widely used in deep learning [37]. The activation function that controlled the non-linearity of individual neurons was Relu, which is widely recommended [17]. There are two hidden layers in the present model. The number of neurons in the hidden layers that control the representational capacity of the network were set to 300, after tuning.

### 2.2.8. Adaptive Neuro-Fuzzy Inference System

Adaptive Neuro-Fuzzy Inference System (ANFIS) models combine fuzzy systems and the learning ability of neural networks. ANFIS is considered an ANN model that conducts the preprocessing step by converting numeric values into fuzzy values [5,32,41–43].

The toolbox feature of the ANFIS forms a fuzzy inference system whose membership structure or parameters can be calibrated either using a backpropagation method alone or combining with the least-squares-type method. To create an inference system, five different layers, namely the fuzzy layer, product layer, normalized layer, de-fuzzy layer, and the total output layer, are used. Each layer consists of different adaptive nodes that exert changeable and fixed factors on input values.

The ANFIS rules are presented in the following form:

$$y = px_1 + qx_2 + r \tag{8}$$

where $x_1, x_2$ are input variables from corresponding fuzzy sets, $y$ is the output, and $p, q, r$ are constant parameters.

### 2.3. Methodological Overview

The collected data were randomly split into two parts. The first part, including 65% of the data, was used for the training phase, while the remaining 35% of the data were are used as the testing set.

In the splitting part of the AI models, the dataset was divided to randomly train and test subsets. This means that data were shuffled first, then 65% of shuffled data were used for training phase, and the remaining data were set aside as testing data. The considered AI approach was fitted to the train subset, and the accuracy of training step was calculated based on several error measures, as described in Equations (9)–(14). In the next step, the fitted AI model was applied to the test subset for prediction purpose. Again, the accuracy of the testing step was calculated.

Different explanations can be made for the applied AI models considering the accuracy of training and testing steps. If the model performs much better on the training set than on the test set, then we are likely to be overfitting. Underfitting refers to a model that can neither model the training data nor the testing data. The sweet spot between underfitting and overfitting, which shows the good performance of a machine learning algorithm on both the training and testing data, is a good fit [5,37].

The accuracy results of the training and testing stages of each applied AI model are presented in Table 3. The values show good performance for all models on both training and testing data, which proves the dataset is not prone to overfitting.

**Table 3.** Checking overfitting and cross validation.

| Accuracy Score | Linear | Lasso | Ridge | Enet | KNN | RF | Gradient Boosting | XG Boost | SVM | Stacking | MLP | Deep Learning | ANFIS |
|---|---|---|---|---|---|---|---|---|---|---|---|---|---|
| Training $R^2$ | 0.96 | 0.96 | 0.96 | 0.96 | 0.96 | 0.99 | 0.99 | 0.99 | 0.99 | 0.98 | 0.98 | 0.98 | 0.96 |
| Testing $R^2$ | 0.94 | 0.94 | 0.94 | 0.94 | 0.91 | 0.94 | 0.95 | 0.94 | 0.97 | 0.96 | 0.96 | 0.97 | 0.94 |
| Mean K-fold CV | 0.95 | 0.95 | 0.95 | 0.95 | 0.89 | 0.95 | 0.96 | 0.95 | 0.97 | 0.96 | 0.96 | 0.96 | 0.95 |
| Sdev K-fold CV | 0.01 | 0.01 | 0.01 | 0.01 | 0.03 | 0.02 | 0.01 | 0.02 | 0.01 | 0.01 | 0.01 | 0.01 | 0.01 |

Moreover, to ensure that splitting does not affect the model performance, cross-validation (CV) metrics are employed in the developed codes. In this function, splitting the data was carried out k consecutive times (here a process of 5 times was typically chosen) with different splits each time. The model was fitted and scores were computed for each iteration. The mean and standard deviations of scores are presented in Table 3 for every AI model. The fact that, for all models, the mean is near 1.0 and the standard deviation is negligible shows that random splitting was successful in the cross-validation evaluation and did not lead to overfitting in the considered case.

In fact, the temperature time series are non-stationary since their statistical properties change over time. In the present study, the soil temperature prediction was primarily considered in the hot season. The mean and standard deviations of the all the data are 296 and 4, respectively. Then, the time series were split into four partitions, and each group's mean and standard deviation were calculated. It was found that the statistical parameters of each division do not have noticeable differences from each other. This shows the non-stationary factor of soil temperature time series used in the present study cannot be considered a major issue.

The air temperature, precipitation, surface pressure, evaporation, instantaneous wind speed, dewpoint temperature, solar radiation and thermal radiation are the atmospheric variables used as the inputs of the benchmark algorithm, and the soil temperature at a depth of 0–7 cm underground is the output of the model.

With the aim of making model training less sensitive to the scale of parameters and allowing our models to converge to better weights and, in turn, lead to a more accurate

model, the data were normalized in a way that removed the mean and scaled each variable to unit variance. Scaling occurred independently on each parameter by computing the relevant statistics on the samples in the training set. Mean and standard deviations were then stored to be used on later data using denormalization. The same procedure was employed for testing data before prediction.

The overall flow of the simulation is illustrated in Figure 2.

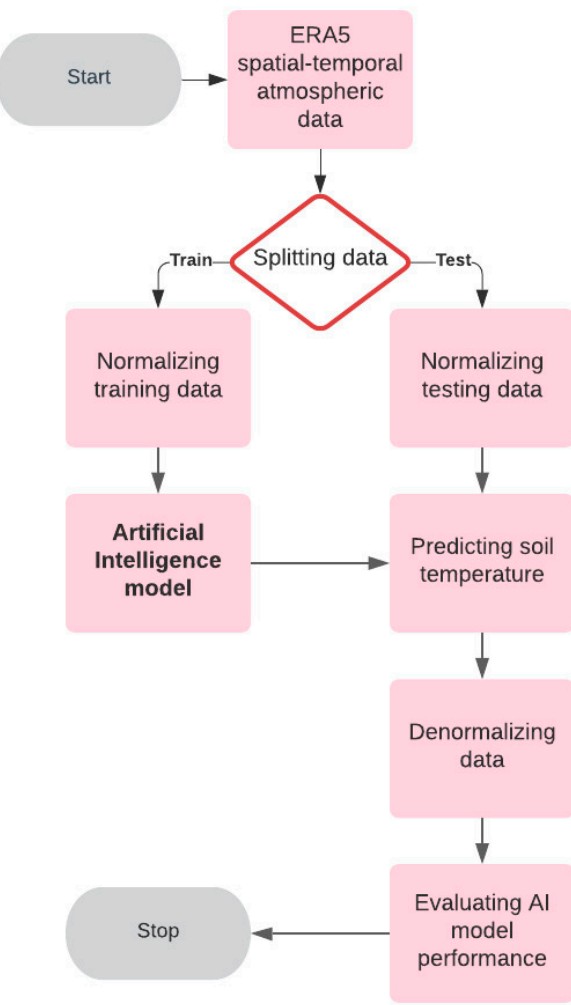

**Figure 2.** Algorithm of the developed numerical model in the present study.

The applied programming language in this study was Python version 3.9 (Python: Open-source, designed by Guido van Rossum in 1991). Python is a high-level, object-oriented, general-purpose, interactive language that is widely used for data analysis and machine learning. Additionally, Spyder version 5.1.5 (Spyder: Open-source, original author is Pierre Raybaut in 2009) was employed, which is a scientific Python development environment. The processor of the used device was 11th Gen Intel Core i5 @ 2.40 GHz, and installed RAM was 8.00 GB.

To facilitate the assessment of the model performance, the outputs of applied AI models need to be compared. Several error indicators are employed to measure the quality of modeling, including maximum residual error (*MaxE*), mean absolute error (*MAE*), mean square error (*MSE*), root mean square error (*RMSE*), normalized root mean square error (*NRMSE*) and coefficient of determination (*R-squared*). The evaluation metrics are defined as:

$$MaxE = Max(y_{obs} - y_{calc}) \quad \text{optimal value: 0} \tag{9}$$

$$MAE = \frac{\sum |y_{obs} - y_{calc}|}{n} \quad \text{optimal value: 0} \tag{10}$$

$$MSE = \frac{\sum (y_{obs} - y_{calc})^2}{n} \quad \text{optimal value: 0} \tag{11}$$

$$RMSE = \sqrt{\frac{\sum (y_{obs} - y_{calc})^2}{n}} \quad \text{optimal value: 0} \tag{12}$$

$$NRMSE = \frac{RMSE}{[Max(y_{obs}) - Min(v_{obs})]} \quad \text{optimal value: 0} \tag{13}$$

$$R^2 = 1 - \frac{\sum (y_{obs} - y_{calc})^2}{\sum (y_{obs} - \overline{y_{calc}})^2} \quad \text{optimal value: 1} \tag{14}$$

where $y_{obs}$ is the observed value, $y_{calc}$ is the predicted value by the AI mode, $\overline{y_{calc}}$ is the mean of calculated values and $n.$ is the number of data points.

## 3. Results

Before applying the models to all stations representing Ottawa, one station (station #2 in Table 1) was selected, and AI models were applied. The location of the station was central–southern Ottawa, with coordinates of 45.25° N and 75.75° W, which are shown in Figure 1c. As mentioned earlier, eight hourly climate parameters were used as input variables. The data were gathered from 1 to 31 July 2020, a total of 31 days, and approximately 6000 pieces of climatic information were found.

According to what was stated, AI models were applied for two sets of data. The first dataset is for a confined database with a limited quantity of information based on station #2 (shown in Figure 1c) and is named as the limited dataset in the current study. The second dataset is an extensive collection of data based on the information of six stations (shown in Figure 1b) and is named as the big dataset in the present study.

After applying each developed AI model on these two datasets, evaluation of the model's performance was carried out separately; then, a comprehensive assessment was finally performed.

The primary step of modeling was splitting the data into two groups. Although the data were randomly split for training and testing purposes, the training and testing data were maintained for all models. So, all AI models were trained and tested with the same set of data. After the model training procedure, the model was fed with testing data as inputs, and prediction results were obtained. The predicted outcomes and real values were simultaneously reshaped into a 1-dimensional series, and the performances of the models were evaluated using the error metrics.

The developed model was executed each time, employing 1 of the 13 abovementioned AI techniques, once on the limited database and once on the big dataset. Hence, 26 sets of predicted data were obtained.

The residuals of soil temperature, the difference between the actual and the predicted values obtained from each AI model were calculated and are illustrated in Figure 3 as box plots. It makes sense that the closer the residual distribution to zero, the better the model prediction.

A box plot is a graphical method for demonstrating the locality and spread of numerical data. This diagram is a standardized way of displaying the five significant summary numbers of the minimum ($Q_0.$), first quartile ($Q_1.$), median ($Q_2.$), third quartile ($Q_3$) and maximum ($Q_4$). Additionally, interquartile range (IQR), the distance between upper and lower quartile ($Q_3 - Q_1$), can be seen from this plot.

Additionally, the graph provides some indication of the symmetry and skewness of the data and shows outliers. Outliers are extreme values that lie at an abnormal distance from other values. Less outliers means that not many points significantly deviate from the rest of the data.

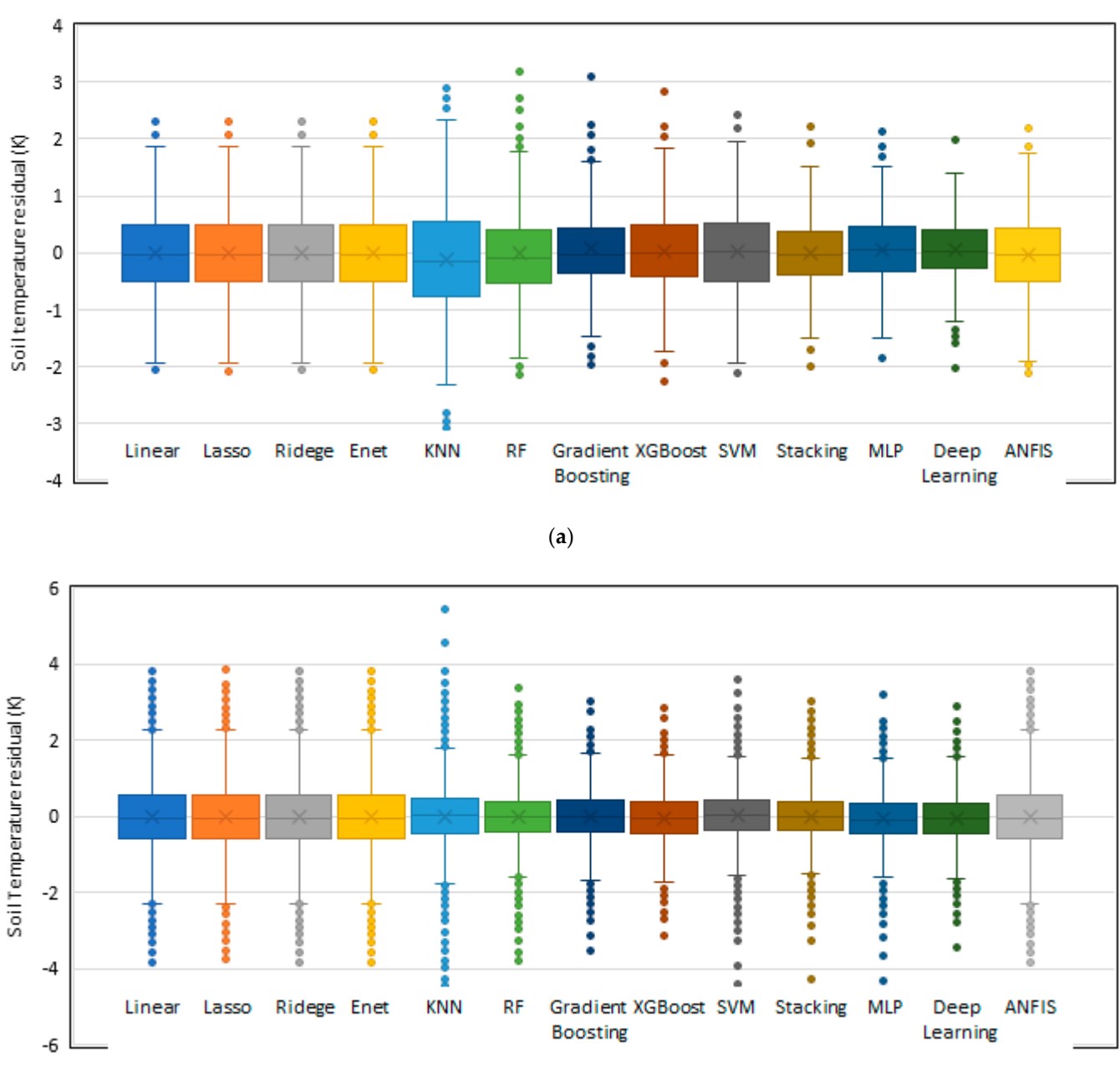

**Figure 3.** Box plots of soil temperature residuals computed for different AI models on (**a**) limited dataset; (**b**) big dataset.

Consequently, investigating the box plots presented in Figure 3 provides helpful information that can be used as an evaluation tool to assess the performances of different AI models in both limited and big datasets.

The closer the median to zero and the smaller the interquartile range, the better the model performance since this indicates that the residuals are mainly distributed around zero. Additionally, fewer outliers are desirable for residuals of an AI model prediction.

Figure 3a demonstrates the box plot of residuals computed by different AI models on the limited dataset. It can be seen from Figure 3a that deep learning residuals have the smallest interquartile range; their median is almost zero, and the absolute value of their maximum and minimum are lower than other models. Therefore, this method has the best performance for predicting soil temperature. Following the deep learning method,

the MLP and stacking methods have the least IQR, lowest $Q_4$, highest $Q_0$ and their $Q_2$ is very close to zero. So, the performance of these two methods is evaluated as effective. On the other hand, the KNN method has the highest IQR, highest $Q_4$, lowest $Q_0$ and its $Q_2$ is clearly less than zero. Thus, this method did not show an acceptable performance in soil temperature prediction.

Figure 3b demonstrates the box plot of residuals computed by different AI models on the big dataset. It can be seen from Figure 3b that the median of all models is very close to zero. The interquartile ranges of deep learning, MLP and stacking methods are less than the others. Deep learning has the lowest maximum and highest minimum among these three methods. Therefore, deep learning followed by MLP and stacking methods also showed the best performance for big dataset.

Figure 4 is a scatter plot of the predicted soil temperatures computed by applying all different AI methods on ERA5 data, and demonstrates a good fit between the observed values and the models' predictions. The predicted soil temperatures show a very close match to the identity line in Figure 4a,b. It was determined that all AI models were able to provide reliable soil temperature results for both the limited dataset and big dataset.

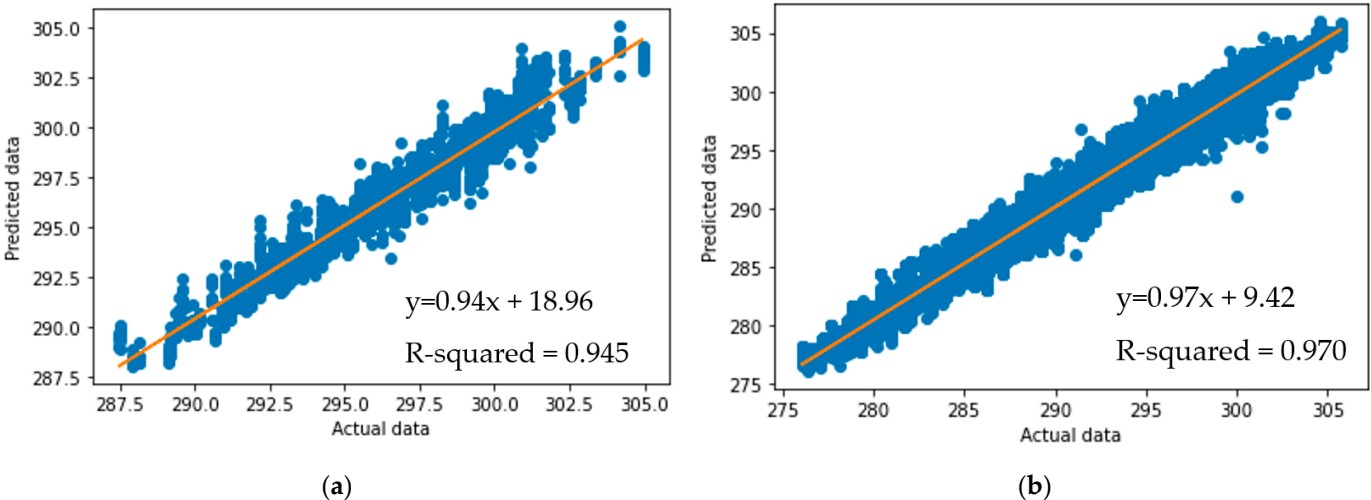

**Figure 4.** Scatter plots of predicted and observed soil temperature using different AI models: (**a**) limited dataset; (**b**) big dataset.

The information presented in Figure 4 shows that the size of data in AI models plays a significant role in the correctness of results, and more data lead to more robust and promising results. The correlation between actual and predicted data was 97% for the big dataset, whereas the R-squared value for the limited dataset was found to be 94.5%.

*Performance of Developed AI Models*

Previous literature reviews show that different researchers employed different AI approaches and developed their models to predict soil temperature. However, among the well-established and advanced AI models, the question of which model is best for making a reasonably precise prediction in a timely manner remains unanswered.

In the current study, thirteen AI models—linear regression, lasso, ridge, Enet, KNN, random forest, gradient boosting, XG boost, support vector machine, stacking, multi-layer perceptron, deep learning and adaptive neuro-fuzzy inference system—were employed to predict soil temperature. The mentioned models were applied to two sets of data with different quantities of information to assess the performance of the various AI models. Meanwhile, the effect of dataset size on the behavior of AI models was evaluated.

To measure the quality of different AI models, the statistical indicators of Equations (9)–(14) were applied, and the results of the error analysis are presented in Table 4 for both limited and big datasets.

**Table 4.** Error analysis of predicted soil temperature values using different AI models.

| Size of Dataset | AI Models | MaxE (K) | MAE (K) | MSE (K²) | RMSE (K) | NRMSE (-) | R² (-) |
|---|---|---|---|---|---|---|---|
| Limited Dataset | Linear | 2.31 | 0.63 | 0.65 | 0.81 | 4.6% | 0.94 |
| | Lasso | 2.31 | 0.63 | 0.65 | 0.81 | 4.6% | 0.94 |
| | Ridge | 2.31 | 0.63 | 0.65 | 0.81 | 4.6% | 0.94 |
| | Enet | 2.31 | 0.63 | 0.65 | 0.81 | 4.6% | 0.94 |
| | KNN | 3.21 | 0.81 | 1.09 | 1.04 | 6.0% | 0.91 |
| | RF | 3.23 | 0.65 | 0.75 | 0.86 | 4.9% | 0.94 |
| | Gradient Boosting | 3.08 | 0.58 | 0.62 | 0.79 | 4.5% | 0.95 |
| | XG Boost | 2.84 | 0.61 | 0.66 | 0.81 | 4.7% | 0.94 |
| | SVM | 2.42 | 0.64 | 0.65 | 0.81 | 4.6% | 0.94 |
| | Stacking | 2.22 | 0.51 | 0.45 | 0.67 | 3.9% | 0.96 |
| | MLP | 2.12 | 0.5 | 0.44 | 0.67 | 3.8% | 0.96 |
| | Deep Learning | 2.04 | 0.46 | 0.39 | 0.62 | 3.6% | 0.97 |
| | ANFIS | 2.2 | 0.62 | 0.65 | 0.81 | 4.6% | 0.94 |
| Big Dataset | Linear | 3.89 | 0.74 | 0.95 | 0.97 | 3.3% | 0.96 |
| | Lasso | 3.90 | 0.74 | 0.95 | 0.98 | 3.3% | 0.96 |
| | Ridge | 3.89 | 0.74 | 0.95 | 0.97 | 3.3% | 0.96 |
| | Enet | 3.89 | 0.74 | 0.95 | 0.97 | 3.3% | 0.96 |
| | KNN | 6.06 | 0.60 | 0.69 | 0.83 | 2.8% | 0.97 |
| | RF | 3.79 | 0.53 | 0.50 | 0.71 | 2.4% | 0.98 |
| | Gradient Boosting | 3.53 | 0.53 | 0.48 | 0.69 | 2.3% | 0.98 |
| | XG Boost | 3.14 | 0.52 | 0.46 | 0.68 | 2.3% | 0.98 |
| | SVM | 4.79 | 0.53 | 0.51 | 0.71 | 2.4% | 0.98 |
| | Stacking | 4.28 | 0.49 | 0.43 | 0.66 | 2.2% | 0.98 |
| | MLP | 4.32 | 0.51 | 0.45 | 0.67 | 2.3% | 0.98 |
| | Deep Learning | 3.44 | 0.51 | 0.44 | 0.66 | 2.2% | 0.98 |
| | ANFIS | 8.99 | 0.74 | 0.97 | 0.98 | 3.3% | 0.96 |

As seen in Table 4, R-squared values are very near 1.00 for both limited and big datasets, which shows a strong correlation between the results predicted by different AI models and soil temperature data. The average calculated R-squared for limited and big datasets equals 0.94 and 0.97, respectively. This outcome is confirmed by the scatter plots illustrated in Figure 4 and offers an overall acceptable performance for all AI methods.

Table 4 indicates that, while employing one AI model using two sets of input data, the AI model works better while increasing the quantity of information, leading to a more robust match between the predicted results and soil temperatures. This conclusion is valid for all applied AI models, as shown in Table 4.

An examination of the error values presented in Table 4 shows that the average NRMSE for the limited dataset was 4.5%, while this value equals 2.7% for the big dataset. This proves that using more data significantly improves error measures, and regardless of which AI method is applied, employing more data leads to better results.

Moreover, as shown in Table 4, four models of linear regression, lasso, ridge and Enet, which have linear bases, showed the same error values for all evaluation metrics in both datasets, demonstrating that they had very similar performances.

As previously mentioned in the methodology section, the last three methods have linear bases and are refined versions of classic linear regression due to the addition of regularization terms. Relatively poor MAE, MSE and NRMSE obtained by the linear regression results demonstrated that this method cannot precisely predict soil temperature. At the same time, the same values of MAE, MSE and NRMSE obtained by lasso, ridge and Enet models showed that linear regression modifications were still not appropriate tools for predicting soil temperatures.

The KNN method had the lowest performance among the investigated AI models for the limited database with the greatest MAE, RMSE and NRMSE. Although this method did not show results with the highest error for the big database, it had one of the lowest

performances among all the considered AI models, with the greatest maximum error and a high RMSE. This finding returns to the logic behind the KNN method, which does not work in the present study. In this method, a number of nearest neighbors are involved in the model, which does not apply for this prediction since it was more temporal than spatial.

A closer look at the error values presented in Table 4 shows that three AI methods: stacking, MLP and deep learning had a better performance than other models. On the limited dataset, the average MAE for these three AI models is less than 0.5 K, while other AI models had MAE of more than 0.65 K. The situation is the same for the big database. The average MAE for these three AI models is approximately 0.5 K, while other AI models had MAE values of approximately 0.65 K. The other error indicator, RMSE, showed a similar trend. On both datasets, the average RMSE for these three AI models was approximately 0.65 K, while other AI models had a RMSE of approximately 0.85 K.

Although deep learning was the best model, the stacking method, which is an ensemble of a few unadvanced models, showed a good performance and predicted the soil temperature with acceptable precision. This performance was better for the big dataset.

It is worth mentioning that the computation cost should be noted as an essential parameter in picking the best method. The execution time for the limited dataset was negligible, but it was significant for the big pieces of information. The average computation time for the deep learning model was 17.5 s, while these values were 10.5 s and 20.5 s for the MLP and stacking models, respectively. The stacking model suffers from insufficient execution speed, despite showing adequate error metrics.

In the statistical analysis, two concepts of the confidence region and prediction bands are often used. The confidence band represents the uncertainty in an estimate of the regression on the data. The prediction band is the region that contains approximately 95% of the points. If another pair of the actual value-calculated value is taken, there is a 95% chance it falls within the prediction band.

The 95% confidence region and 95% prediction band for these three models—stacking, MLP and deep learning—on both limited and big datasets are depicted in Figure 5. The confidence bands in Figure 5 support the previously mentioned claim regarding the strength of these AI models.

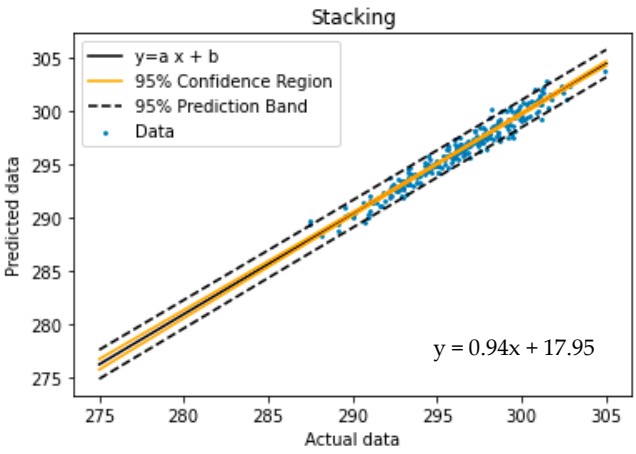
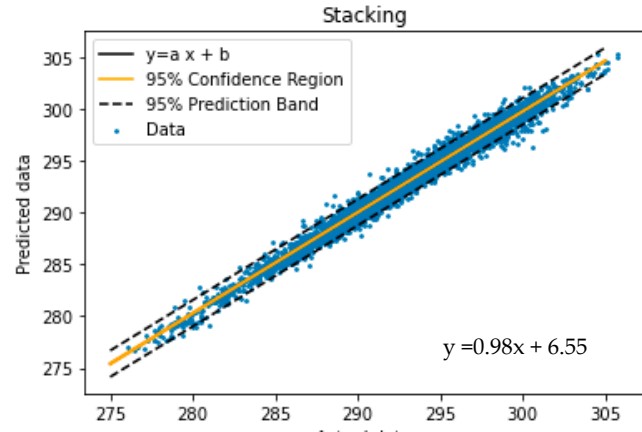

**Figure 5.** *Cont.*

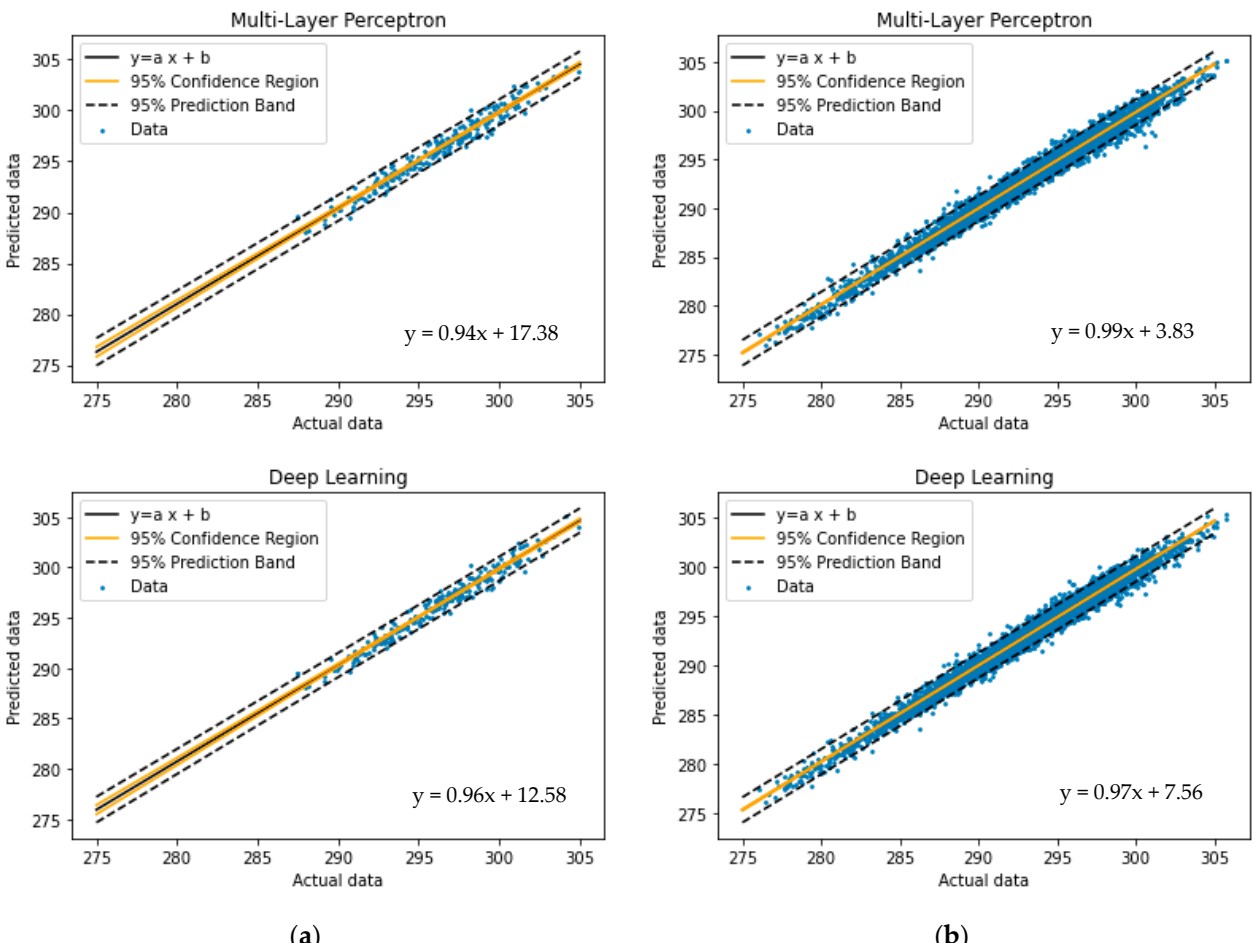

**Figure 5.** Confidence regions and prediction bands of different AI models (**a**) limited dataset (**b**) big dataset.

## 4. Discussion

### 4.1. Importance of Different Input Variables

As previously described, this study aims to fill one main gap in the literature: determining the most important atmospheric variables to be used as input data for AI models. The literature reviews on soil temperature prediction papers published in the last 10 years show that mostly air temperature was used as the only input parameter in the models.

Although apart from air temperature, other different parameters are involved in the models of published papers, the participation percentage was not that high. In papers published in the last 10 years, the rough participation rate of climate variables as input parameters in soil temperature prediction models is as follows [5,7,8,17,29,35–39,41]: Solar radiation and humidity are involved in 40% of studies; wind and pressure are considered in 25% of papers; and less than 15% of studies explored rainfall, sunshine and dewpoint. Additionally, 30% of the models used monthly parameters and 55% considered parameters on a daily basis.

This information shows that, considering eight climate parameters, including air temperature, precipitation, pressure, evaporation, wind, dewpoint, solar radiation, and thermal radiation, on an hourly basis was not typical in previous investigations. The strategy of this study was to consider as many related parameters as possible, finding the most relevant ones by sensitivity analysis. Following this argument, irrelevant parameters should be disregarded in future studies.

A sensitivity analysis was conducted to determine the importance level and relevance rate of atmospheric parameters used as inputs for the AI models. In this regard, the three selected AI models of stacking, MLP and deep learning, which had better performances,

were executed several times. For each run, one of the input parameters was omitted. Then, the code was implemented with the remaining seven variables out of eight preliminary inputs of air temperature, precipitation, surface pressure, evaporation, wind, dewpoint temperature, solar radiation and thermal radiation. The results for the error indicators of the AI model with the remaining seven input parameters are presented in Table 5.

**Table 5.** Sensitivity analysis of input variables for AI models.

| AI Model | Omitted Variable | MaxE | MAE | MSE | RMSE | NRMSE | $R^2$ | Importance |
|---|---|---|---|---|---|---|---|---|
| Stacking, Limited data | precipitation | 2.22 | 0.50 | 0.44 | 0.66 | 3.8% | 0.96 | 8 |
| | pressure | 2.39 | 0.56 | 0.51 | 0.72 | 4.1% | 0.96 | |
| | evaporation | 2.29 | 0.56 | 0.56 | 0.75 | 4.3% | 0.95 | |
| | wind | 2.16 | 0.53 | 0.48 | 0.69 | 3.9% | 0.96 | |
| | dewpoint | 2.22 | 0.52 | 0.48 | 0.69 | 4.0% | 0.96 | |
| | solar radiation | 2.43 | 0.60 | 0.61 | 0.78 | 4.5% | 0.95 | 2 |
| | thermal radiation | 2.12 | 0.52 | 0.46 | 0.68 | 3.9% | 0.96 | |
| | air temperature | 7.31 | 1.39 | 3.24 | 1.80 | 10.3% | 0.72 | 1 |
| MLP, Limited data | precipitation | 1.98 | 0.49 | 0.42 | 0.65 | 3.7% | 0.96 | 8 |
| | pressure | 2.17 | 0.53 | 0.46 | 0.68 | 3.9% | 0.96 | |
| | evaporation | 2.32 | 0.54 | 0.52 | 0.72 | 4.1% | 0.95 | |
| | wind | 2.12 | 0.53 | 0.47 | 0.69 | 3.9% | 0.96 | |
| | dewpoint | 1.93 | 0.50 | 0.45 | 0.67 | 3.8% | 0.96 | |
| | solar radiation | 2.21 | 0.58 | 0.59 | 0.77 | 4.4% | 0.95 | 2 |
| | thermal radiation | 2.19 | 0.52 | 0.46 | 0.68 | 3.9% | 0.96 | |
| | air temperature | 5.92 | 1.20 | 2.50 | 1.58 | 9.0% | 0.79 | 1 |
| Deep Learning, Limited data | precipitation | 2.09 | 0.50 | 0.43 | 0.66 | 3.8% | 0.96 | 7 |
| | pressure | 2.18 | 0.52 | 0.45 | 0.67 | 3.9% | 0.96 | |
| | evaporation | 2.16 | 0.52 | 0.49 | 0.70 | 4.0% | 0.96 | |
| | wind | 2.01 | 0.53 | 0.50 | 0.70 | 4.0% | 0.96 | |
| | dewpoint | 2.27 | 0.49 | 0.45 | 0.67 | 3.9% | 0.96 | |
| | solar radiation | 2.51 | 0.59 | 0.61 | 0.78 | 4.5% | 0.95 | 2 |
| | thermal radiation | 2.29 | 0.49 | 0.42 | 0.65 | 3.7% | 0.96 | 8 |
| | air temperature | 5.69 | 1.33 | 2.93 | 1.71 | 9.8% | 0.75 | 1 |
| Stacking, Big data | precipitation | 4.16 | 0.49 | 0.43 | 0.66 | 2.2% | 0.98 | 8 |
| | pressure | 4.21 | 0.52 | 0.47 | 0.68 | 2.3% | 0.98 | |
| | evaporation | 3.76 | 0.54 | 0.54 | 0.73 | 2.5% | 0.98 | |
| | wind | 3.80 | 0.54 | 0.53 | 0.73 | 2.4% | 0.98 | |
| | dewpoint | 3.99 | 0.53 | 0.51 | 0.71 | 2.4% | 0.98 | |
| | solar radiation | 4.23 | 0.59 | 0.64 | 0.80 | 2.7% | 0.97 | 2 |
| | thermal radiation | 4.26 | 0.52 | 0.49 | 0.70 | 2.4% | 0.98 | |
| | air temperature | 10.24 | 1.19 | 2.78 | 1.67 | 5.6% | 0.88 | 1 |
| MLP, Big data | precipitation | 3.82 | 0.53 | 0.49 | 0.70 | 2.3% | 0.98 | 8 |
| | pressure | 4.27 | 0.53 | 0.49 | 0.70 | 2.4% | 0.98 | |
| | evaporation | 4.10 | 0.60 | 0.64 | 0.80 | 2.7% | 0.97 | |
| | wind | 3.88 | 0.58 | 0.59 | 0.77 | 2.6% | 0.97 | |
| | dewpoint | 3.82 | 0.59 | 0.62 | 0.79 | 2.6% | 0.97 | |
| | solar radiation | 4.34 | 0.65 | 0.74 | 0.86 | 2.9% | 0.97 | 2 |
| | thermal radiation | 4.13 | 0.56 | 0.55 | 0.74 | 2.5% | 0.98 | |
| | air temperature | 10.50 | 1.32 | 3.19 | 1.78 | 6.0% | 0.86 | 1 |
| Deep Learning, Big data | precipitation | 3.94 | 0.52 | 0.46 | 0.68 | 2.3% | 0.98 | 8 |
| | pressure | 4.33 | 0.52 | 0.46 | 0.68 | 2.3% | 0.98 | |
| | evaporation | 3.74 | 0.57 | 0.58 | 0.76 | 2.6% | 0.97 | |
| | wind | 3.71 | 0.58 | 0.58 | 0.76 | 2.6% | 0.97 | |
| | dewpoint | 3.58 | 0.57 | 0.54 | 0.73 | 2.5% | 0.98 | |
| | solar radiation | 4.43 | 0.61 | 0.65 | 0.81 | 2.7% | 0.97 | 2 |
| | thermal radiation | 3.84 | 0.52 | 0.48 | 0.69 | 2.3% | 0.98 | |
| | air temperature | 10.03 | 1.28 | 2.96 | 1.72 | 5.8% | 0.87 | 1 |



Table 5 shows that omitting air temperature from the input variables of the AI model leads to a low correlation coefficient and high NRMSE. It can be concluded that air temperature is a very relevant variable that is highly important for soil temperature prediction.

The same trend occurred for all three AI models and both limited and big datasets applied in the sensitivity analysis.

The soil layer whose temperature is investigated in the present study is a near-surface layer with a depth of 7 cm underground. The air temperature data used in the present study are air temperature at 2 m above the ground surface. Naturally, this soil layer directly touches the air near the ground surface. So, it is logical that the soil temperature is mainly affected by the air temperature.

When comparing the obtained errors with the original error values and considering all eight input parameters, as presented in Table 4, solar radiation is the most critical variable in soil temperature prediction after air temperature. The next most important variable is evaporation.

Additionally, sensitivity analysis displays that precipitation has a negligible effect on results. So, the precipitation does not play an important role in soil temperature forecast and can be omitted from the prediction models without decreasing precision. The mentioned importance level is the same for all three AI models and in both datasets. The level of importance and relevance of each variable is introduced in the last column.

Another investigation on the effects of individual input parameters on soil temperature prediction was carried out. The correlation of soil temperature computed by the MLP model vs. actual data is calculated in Table 6. Additionally, the scatter plots of predicted results and actual data are demonstrated in Figure 6. The fit lines in Figure 6 scatter plots represent the effect of each individual input climate parameters in soil temperature prediction.

The R-squared presented in Table 6 shows that air temperature and precipitation have the highest and smallest correlation, respectively. This finding confirms that air temperature is the most relevant variable in soil temperature prediction and precipitation can be removed from the input set without significant changes in the accuracy of the results.

**Table 6.** Effects of individual input parameters on soil temperature prediction.

| Input Variable | $R^2$ |
|---|---|
| all 8 parameters | 96.7% |
| precipitation | 1.07% |
| pressure | 10.9% |
| evaporation | 34.5% |
| wind | 24.1% |
| dewpoint | 19.9% |
| solar radiation | 28.5% |
| thermal radiation | 13.3% |
| air temperature | 91.7% |

*4.2. Performance of Developed AI Models in Extreme Heat Events*

Since hot warnings are important forecasts used to protect lives and properties, it is essential to ensure the developed prediction models work properly in extremely hot conditions as well. So, the performance of developed AI models in soil temperature prediction has been evaluated in extreme heat events. The model results were investigated to assess whether the models that worked appropriately in ordinary conditions can precisely forecast soil variables in hot weather.

Several studies used several thresholds (from 90% to 99% of the normal distribution) to distinguish extreme events from ordinary ones. Some researchers employed a threshold value based on their knowledge from the region that this value can cause severe consequences. There is no unique definition for extreme threshold in the literature [24].

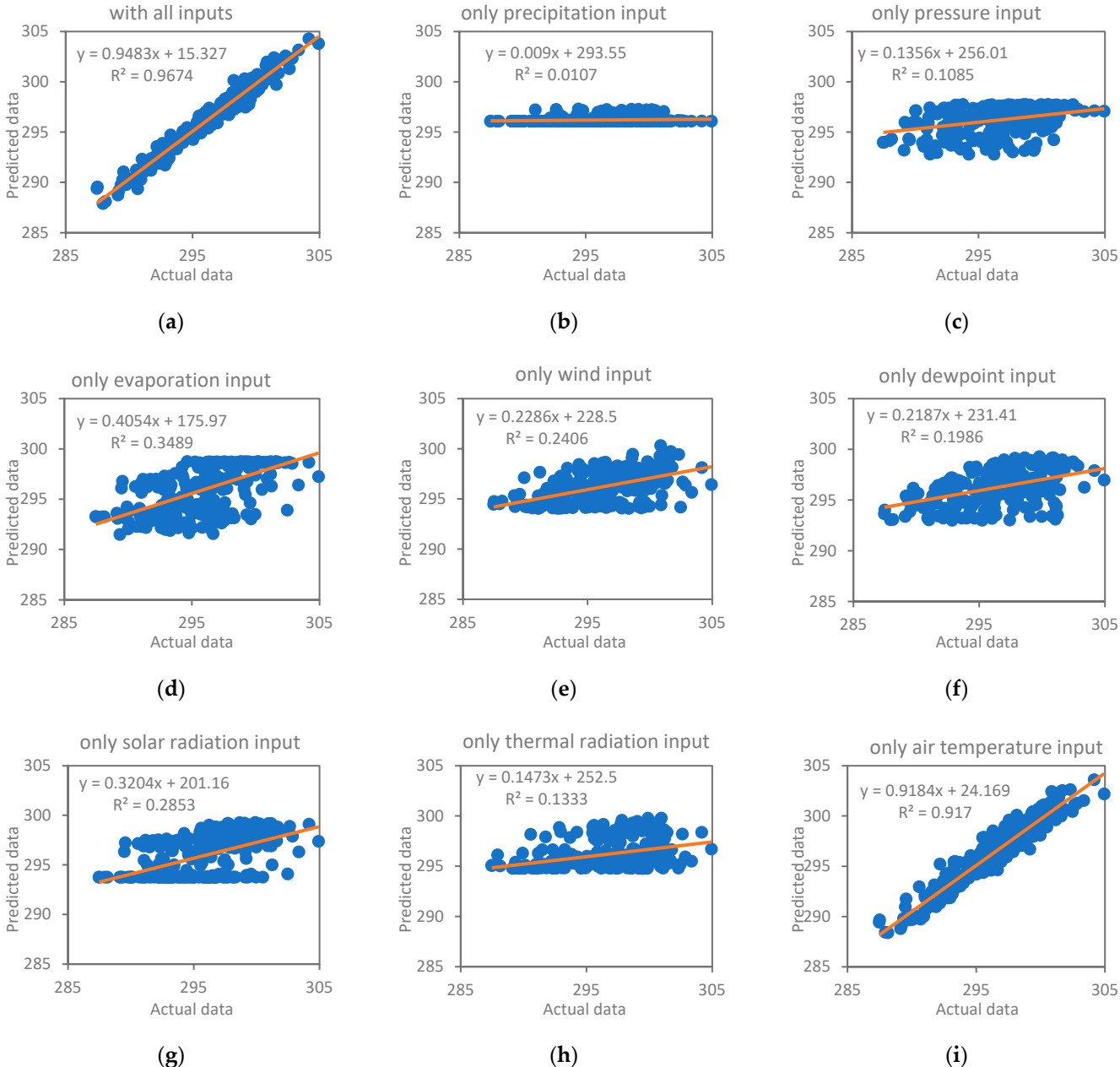

**Figure 6.** Scatter plots of individual input parameters on soil temperature prediction versus actual data (**a**) with all inputs (**b**) only precipitation (**c**) only pressure (**d**) only evaporation (**e**) only wind (**f**) only dewpoint (**g**) only solar radiation (**h**) only thermal radiation (**i**) only air temperature.

Herein, to find the extreme heat events, the soil temperature data was sorted in descending order, for both limited and big datasets separately. The upper decile of data would be labelled as extreme events in the considered case. As mentioned in Section 3, the datasets belong to the summer months of 2020. Thus, the upper decile of data is 10 percent of higher temperatures and are the extreme temperatures that soil has experienced during the considered period. Doing that, there would be 74 and 1325 extreme events in limited and big datasets, respectively.

The developed AI models were applied to the upper decile for both limited and big datasets, and soil temperatures in extreme heat events were predicted. The calculated error indicators are presented in Table 7.

**Table 7.** Error analysis of predicted soil temperature in extremely hot events using different AI models.

| Size of Dataset | AI Models | MaxE (K) | MAE (K) | MSE (K$^2$) | RMSE (K) | NRMSE (-) | R$^2$ (-) |
|---|---|---|---|---|---|---|---|
| Limited Dataset | Linear | 2.50 | 0.70 | 0.75 | 0.87 | 13.2% | 0.67 |
| | Lasso | 2.50 | 0.70 | 0.75 | 0.87 | 13.2% | 0.67 |
| | Ridge | 2.50 | 0.70 | 0.75 | 0.87 | 13.2% | 0.67 |
| | Enet | 2.50 | 0.70 | 0.75 | 0.87 | 13.2% | 0.67 |
| | KNN | 1.90 | 0.71 | 0.75 | 0.86 | 13.2% | 0.67 |
| | RF | 2.31 | 0.62 | 0.63 | 0.79 | 12.1% | 0.72 |
| | Gradient Boosting | 2.44 | 0.65 | 0.70 | 0.84 | 12.8% | 0.69 |
| | XG Boost | 1.79 | 0.68 | 0.74 | 0.86 | 13.1% | 0.68 |
| | SVM | 1.86 | 0.55 | 0.52 | 0.72 | 11.0% | 0.77 |
| | Stacking | 2.10 | 0.56 | 0.52 | 0.72 | 11.0% | 0.77 |
| | MLP | 2.12 | 0.52 | 0.45 | 0.67 | 10.2% | 0.80 |
| | Deep Learning | 2.06 | 0.51 | 0.44 | 0.66 | 10.1% | 0.81 |
| Big Dataset | Linear | 3.42 | 0.67 | 0.73 | 0.86 | 8.7% | 0.76 |
| | Lasso | 3.42 | 0.67 | 0.73 | 0.86 | 8.7% | 0.76 |
| | Ridge | 3.42 | 0.67 | 0.73 | 0.86 | 8.7% | 0.76 |
| | Enet | 3.42 | 0.67 | 0.73 | 0.86 | 8.7% | 0.76 |
| | KNN | 3.34 | 0.56 | 0.55 | 0.74 | 7.5% | 0.82 |
| | RF | 4.12 | 0.55 | 0.52 | 0.72 | 7.3% | 0.83 |
| | Gradient Boosting | 4.37 | 0.56 | 0.53 | 0.73 | 7.4% | 0.82 |
| | XG Boost | 4.23 | 0.54 | 0.51 | 0.72 | 7.3% | 0.83 |
| | SVM | 3.57 | 0.52 | 0.47 | 0.69 | 7.0% | 0.84 |
| | Stacking | 3.96 | 0.52 | 0.48 | 0.69 | 7.0% | 0.84 |
| | MLP | 3.01 | 0.50 | 0.42 | 0.65 | 6.6% | 0.86 |
| | Deep Learning | 3.01 | 0.49 | 0.41 | 0.64 | 6.5% | 0.86 |

It can be seen from Table 7 that from classic regressions to well-established methods to advanced approaches, error indicators have improved. This indicates that predicted results would improve significantly using more advanced AI models like SVM, stacking, MLP and deep learning. Among all employed AI approaches, deep learning and MLP showed the best performance since they had the highest R-squared and the lowest NRMSE. The same pattern can be recognized for the big dataset, in which the two methods of deep learning and MLP presented the finest error indexes. This finding confirms that deep learning followed by MLP methods is not only the best approach for predicting soil temperature in ordinary climate conditions, but also shows robust tools to forecast soil temperature in extremely hot weather.

To better understand the ability of developed AI models for extremely hot weather predictions and compare these outcomes with ordinary climate conditions, soil temperature prediction results for both circumstances are presented in Figure 7. Figure 7b illustrates the NRMSE and R-squared calculated in extreme heat, while Figure 7a demonstrates the same parameters for ordinary conditions. These two graphs reveal that prediction in extreme conditions leads to lower correlation coefficients and higher error for all AI models and both datasets. Figure 7c displays a closer look at NRMSE changes for extreme events and ordinary conditions. It can be seen that, although the error indicator of ordinary conditions is less than for extreme events, advanced models such as deep learning and MLP predict the soil temperature in extremely hot weather with appropriate precision. Similarly, the R-squared value presented in Figure 7d for extreme events and ordinary conditions indicate that the soil temperature predictions in extremely hot weather could reach an acceptable value through advanced models such as deep learning and MLP. However, the R-squared values are lower than those related to ordinary conditions.

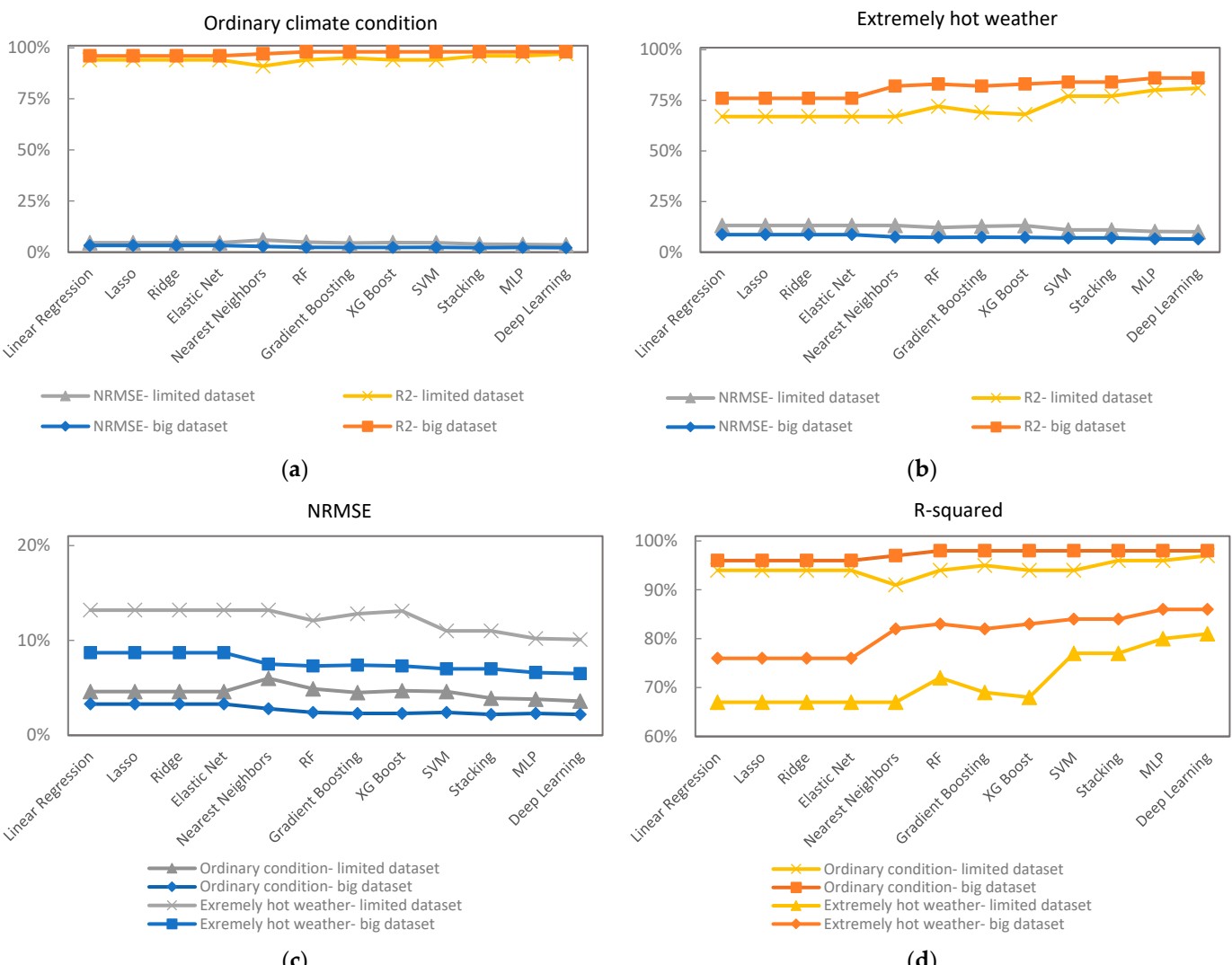

**Figure 7.** Comparison of error index of predicted soil temperature in ordinary climate conditions and extremely hot events for both limited and big datasets: (**a**) error index in ordinary climate condition; (**b**) error index in extremely hot events; (**c**) NRMSE comparison; and (**d**) R-squared comparison.

### 4.3. Importance of Different Input Variables in Extreme Heat Events

Similar to the experiments in ordinary weather conditions, a sensitivity analysis was conducted on AI prediction models applied to extreme heat data to realize the importance level of each climate variable that was involved. This analysis was based on omitting one input parameter and investigating the model performance in the absence of that parameter. The calculated results are presented in Table 8.

The error metrics in Table 8 show that the prediction of hot weather confirmed the earlier findings on the importance of air temperature and solar radiation as levels one and two in soil temperature prediction in extreme heat events. Additionally, Table 8 indicates that precipitation has only a slight relevance in soil temperature prediction in extremely hot conditions.

**Table 8.** Sensitivity analysis of input variables in extremely hot events.

| AI Model | Omitted Variable | MaxE | MAE | MSE | RMSE | NRMSE | $R^2$ | Importance |
|---|---|---|---|---|---|---|---|---|
| | precipitation | 2.20 | 0.53 | 0.46 | 0.68 | 10.4% | 0.80 | 7 |
| | pressure | 2.22 | 0.52 | 0.48 | 0.69 | 10.5% | 0.79 | |
| | evaporation | 2.11 | 0.51 | 0.42 | 0.64 | 9.8% | 0.82 | 8 |
| MLP, Limited Data, | wind | 2.38 | 0.53 | 0.46 | 0.68 | 10.4% | 0.80 | |
| extremely hot events | dewpoint | 2.09 | 0.53 | 0.48 | 0.69 | 10.6% | 0.79 | |
| | solar radiation | 2.13 | 0.56 | 0.53 | 0.73 | 11.1% | 0.77 | 2 |
| | thermal radiation | 2.05 | 0.55 | 0.50 | 0.71 | 10.8% | 0.78 | |
| | air temperature | 3.71 | 0.99 | 1.55 | 1.24 | 19.0% | 0.32 | 1 |
| | precipitation | 3.61 | 0.54 | 0.48 | 0.69 | 7.0% | 0.84 | 8 |
| | pressure | 3.24 | 0.54 | 0.49 | 0.70 | 7.1% | 0.84 | |
| | evaporation | 3.72 | 0.54 | 0.50 | 0.71 | 7.2% | 0.83 | |
| MLP, Big Data, | wind | 3.72 | 0.53 | 0.50 | 0.70 | 7.1% | 0.83 | |
| extremely hot events | dewpoint | 3.36 | 0.57 | 0.55 | 0.74 | 7.5% | 0.82 | |
| | solar radiation | 3.33 | 0.57 | 0.56 | 0.75 | 7.6% | 0.81 | 2 |
| | thermal radiation | 3.08 | 0.53 | 0.47 | 0.69 | 7.0% | 0.84 | |
| | air temperature | 7.41 | 1.08 | 1.91 | 1.38 | 14.0% | 0.36 | 1 |

## 5. Conclusions

A precise and cost-effective model for soil temperature forecasting, which has the advantages of artificial intelligence techniques, is developed in the present research. Therefore, 13 AI models—linear regression, ridge, lasso, Enet, KNN, RF, gradient boosting, XG boosting, stacking method, SVM, MLP, deep learning and ANFIS—were employed to generate a comprehensive and detailed assessment of the performance of different AI approaches in soil temperature estimation. In this regard, eight hourly land and atmospheric variables of air temperature, precipitation, surface pressure, evaporation, wind gust, dewpoint temperature, solar radiation and thermal radiation were employed, and predictions were made using two limited and big datasets. The results show that AI is a promising approach in climate parameter forecast, and developed AI models show a reliable ability in soil temperature prediction. Additionally, applying AI models to more information led to better results, even when using the same method.

The key findings of this study are summarized as follows:

- Among all 13 AI models applied in the current study, deep learning, followed by the MLP method, showed the best performance in predicting soil temperature with the highest correlation coefficient and lowest error metrics.
- Although deep learning was the best model, the stacking method showed a good performance with an acceptable precision in soil temperature prediction.
- A sensitivity analysis shows that air temperature and solar radiation play the most important roles in soil temperature prediction, while precipitation can be neglected in forecast AI models.
- The evaluation of developed AI models in hot weather confirmed the most successful performance of deep learning and MLP methods in extreme heat events compared to other employed models.
- The results of extreme heat events show a moderate decrease in performance compared to the models' outcomes for the ordinary weather conditions. This reduction is more meaningful for classic regression rather than advanced AI models.
- A sensitivity analysis of involved variables for predictions in hot weather confirmed the earlier findings of the importance level of air temperature, solar radiation and precipitation in soil temperature prediction in extreme hot events.

**Author Contributions:** Conceptualization, J.H.C., A.M., H.S. and P.P.; methodology, H.I.; software, H.I.; validation, H.I. and P.P.; formal analysis, H.I. and P.P.; investigation, H.I.; resources, H.I.; data curation, H.I.; writing—original draft preparation, H.I. and A.M.; writing—review and editing J.H.C., P.P., H.S. and A.M.; visualization, H.I.; supervision, A.M. and J.H.C.; project administration, A.M., J.H.C. and H.S.; funding acquisition, A.M., H.S. and J.H.C. All authors have read and agreed to the published version of the manuscript.

**Funding:** This research was funded by National Research Council Canada through the Artificial Intelligence for Logistics Supercluster Support Program, grant number AI4L-120.

**Data Availability Statement:** Parts of the data used in this manuscript are available through the corresponding author upon reasonable request.

**Conflicts of Interest:** The authors declare no conflict of interest.

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
