# Peer review of "A Comprehensive Study of Artificial Intelligence Applications for Soil Temperature Prediction in Ordinary Climate Conditions and Extremely Hot Events"

_sustainability, doi:10.3390/su14138065_

Round 1

Reviewer 1 Report

General comments:

This study selected thirteen approaches from AI techniques to predict soil temperature, and evaluated the sensitivity of different climate variables in predicting the soil temperature. This study is an important and meaningful study.But this manuscript need major revisions before they can be published.

Special comments:

1. Line32, Too many keywords, should be reduced to 5 or 6.

2. Line35-172, Some paragraphs are too short and need to be integrated; and many examples are given, lacking a summary.

3. Line190, Latitude and longitude are missing in Figure 1.

4. Line392, It is suggested that the names of the 13 methods should be uniformly abbreviated, rather than some abbreviations and some not.

5. There is a lot of content in the Discussion section that belongs to Results, and the two parts of Results and Discussion can be combined into Results and Discussion.

6. Line587, There is no black line in the figure, and the specific regression equation needs to be given, that is, the parameter values of a and b, rather than y=ax+b directly.

Author Response

Besides, the manuscript is edited by a native English speaker and professional proofreader with 11 years of experience.

Reviewer 2 Report

The text of the manuscript is written in good scientific style, the presentation of the material is consistent, the structure of the manuscript is logical. However, some things in the manuscript are not explained in great detail. And I have a number of comments:

lines 196-200. I did not see information about where the soil temperature data was taken from?

line 222. Error in the designations.

line 245. It is not clear from Table 2 how much R-Squared changed at large alpha values ​​(eg 1E-2). Maybe it changes just as little?

Section 2.2. few references to the literature.

line 321. Is the ReLU function always used in the MLP?

lines 340-350. You write that deep learning includes large neural networks, but in this example, as I understand from the text, you use only one hidden layer? This is much less than for the MLP above. How appropriate is this?

lines 393-396. The authors should give comments on taking into account the nonstationarity factor of the temperature field. After all, the soil at a depth of 7 cm does not instantly respond to changes in air temperature.

line 515. I didn't see a match between the R-squared values ​​in table 3 and table 4. Should there be some match between these values?

lines 615-618. The authors should give comments from the point of view of physics why air temperature affects the most, while other parameters affect less.

line 619. Judging by Table 6, the second most important parameter here is evaporation. But in Table 5, it has a high R-squared value and was not highlighted as significant. With what it can be connected?

Round 2

Reviewer 1 Report

The author addresses my concerns with no further comments.

Reviewer 2 Report

The authors did a good job on the manuscript and significantly improved it. Now I have no comments.